# Surgical Treatment for Early Cervical Cancer in the HPV Era: State of the Art

**DOI:** 10.3390/healthcare11222942

**Published:** 2023-11-10

**Authors:** Mario Palumbo, Luigi Della Corte, Carlo Ronsini, Serena Guerra, Pierluigi Giampaolino, Giuseppe Bifulco

**Affiliations:** 1Department of Public Health, School of Medicine, University of Naples “Federico II”, 80131 Naples, Italy; mpalumbomed@gmail.com (M.P.); sere.guerra@gmail.com (S.G.); pgiampaolino@gmail.com (P.G.); giuseppe.bifulco@unina.it (G.B.); 2Department of Neuroscience, Reproductive Sciences and Dentistry, School of Medicine, University of Naples “Federico II”, 80131 Naples, Italy; 3Department of Woman, Child and General and Specialized Surgery, School of Medicine, University of Campania “Luigi Vanvitelli”, 80138 Naples, Italy; carlo.ronsini90@gmail.com

**Keywords:** HPV, early cervical cancer, cervical dysplasia, surgical treatment

## Abstract

Cervical cancer (CC) is the fourth most common cancer among women worldwide. The aim of this study is to focus on the state of the art of CC prevention, early diagnosis, and treatment and, within the latter, the role of surgery in the various stages of the disease with a focus on the impact of the LACC study (Laparoscopic Approach to Cervical Cancer trial) on the scientific debate and clinical practice. We have discussed the controversial application of minimally invasive surgery (MIS) for tumors < 2 cm and the possibility of fertility-sparing surgery on young women desirous of pregnancy. This analysis provides support for surgeons in the choice of better management, including patients with a desire for offspring and the need for sentinel node biopsy (SNB) rather than pelvic lymphadenectomy for tumors < 4 cm, and without suspicious lymph nodes’ involvement on imaging. Vaccines and early diagnosis of pre-cancerous lesions are the most effective public health tool to tackle cervical cancer worldwide.

## 1. Introduction

### 1.1. Epidemiology

Cervical cancer (CC) is the fourth most common female malignancy worldwide and represents a major global health problem [1].

In 2020, an estimated 604,127 new cases of cervix uteri neoplasms were diagnosed, and an estimated 341,831 deaths occurred worldwide. In accordance with the literature, it is possible to estimate 14,480 new cases of CC and 4290 deaths from this cancer among women in the United States in 2021 [1,2]. The age-standardised mortality rates account for almost half of the incidence rates (133 and 73 per 100,000, respectively) on a global scale [2]. In-depth analysis demonstrates the mortality-to-incidence ratio (MIR) to be 0.71 in Eastern Africa versus 0.211 in Northern Europe: a two-speed public health challenge [2].

Human papillomavirus (HPV) is a huge family of viruses, and an etiologic agent of the most common sexually transmitted infection in the United States, with highest infection rates in the population aged 20 to 25 years old. In healthy, non-immunocompromised woman, the infection consists of a self-limiting process [2,3].

HPV presents more than 100 serotypes. Among others, HPV types 6 and 11 (low risk serotypes) are responsible for genital warts, whereas types 16 and 18 (high risk serotypes) are causally related to CC [3].

Human papillomavirus infects cells in the basal layer of the epithelium, probably via micro-abrasions in the epithelial surface [4]. Viral replication occurs only in suprabasal, differentiating cells that are destined for maturity and senescence within 2–3 weeks of infection. In the case of malignant transformation, stepwise carcinogenic progression occurs through different grades of dysplasia referred to as Cervical Intraepithelial Neoplasia (CIN) 1, 2, 3. An epithelial basement membrane breach marks carcinoma in situ with progression to invasive cancer [3,4].

Persistent HPV infection remains the leading cause of CC. The virus reaches the basal layer of the epithelium and induces the molecular mechanisms of carcinogenesis linked to P53 disregulation over the years [3]. In low-income countries, vaccination is not routinely performed and screening is based on opportunistic cervical cytology, while in high-income countries, vaccination coverage is adequate and HPV DNA testing is the main test for screening [4].

The estimated HPV prevalence among women with normal cytological findings worldwide is 11.7%. The most effective strategy to prevent the disease is vaccination against HPV. There are three types of HPV vaccines: “Quadrivalent vaccine” (Gardasil: against HPV types 6, 11, 16, 18); “Bivalent vaccine” (Cervarix: against HPV types 16, 18); “9-valent vaccine” (Gardasil 9: against HPV types 6, 11, 16, 18, 31, 33, 45, 52, 58) [5]. In the case of CC diagnosis (FIGO stages IA1 with lymph-vascular space invasion (LVSI), IA2, IB1, IB2, and IIA1), radical hysterectomy (RH) with bilateral pelvic lymphadenectomy is considered the gold standard treatment [5,6].

### 1.2. Cervical Cancer Screening Tests

Observational studies have clearly demonstrated a reduction in invasive cancer incidence and mortality in well-organized screening programs using cervical cytological testing that have been implemented [7]. Also, randomized controlled trials (RCTs) of well-screened populations have shown that strategies incorporating testing for high-risk Human PapillomaVirus (hrHPV) subtypes were, on aggregate, associated with a reduction in invasive cancer incidence relative to that shown by cytological screening alone [8]. Current guidelines recommend three primary screening options: cytological testing alone, hrHPV testing, and cytological + hrHPV combination testing (co-testing) [9,10].

Exfoliative cytology, first described by Papanicolau in the 1940s, has been the mainstay for cervical screening, updated with liquid-based cytology (LBC), proven to reduce the rate of unsatisfactory smears [11,12] and allow molecular testing, compared to conventional cytology (CC). The majority of abnormalities are in squamous cells. Smear analysis grants high specificity and positive predictive value for HG CIN but with high false-negative rates (20–25%) [12,13]. Also, it has limitations in detecting glandular intraepithelial lesions located in endocervical glands [14,15]. Incorrect sampling or interpretation accounts for 30% of new CC cases [16,17].

HPV tests have substantially higher sensitivity and negative predictive value for high-grade disease when compared to cytology. The higher sensitivity permits longer screening intervals from 5 to 10 years [18,19]. Oncogenic HPV screening tests have low inter- and intra-variability, reduces the number of unsatisfactory results, and permits self-sampling. The major disadvantage of hrHPV testing is its very age-dependent lower specificity when compared to cytology, as the test can detect transient HPV infections without a true carcinogenic potential. The use of hrHPV primary screening on women under 30 years of age is not advised, because of the high prevalence of hrHPV infections in this age group [20]. To improve specificity and minimise over-referral to colposcopy, triage tests are needed to identify infections more likely to be persistent and associated with the development of cancer. Triage tests include cellular assays such as cytological findings, p16 with or without Ki67 immunostaining (only on clinician-taken samples), genotyping, viral load, and methylation assays (on either self- or clinician-collected samples) [21,22].

A recent systematic review and meta-analysis by Terasawa et al., including twenty-seven prospective studies, compared cytology and/or high-risk human papillomavirus (hrHPV) screening test performance at detecting cervical intraepithelial neoplasia ≥ grade 2 (CIN2+) in healthy asymptomatic women [23]. Results, congruent with other meta-analysis, found the OR rule (i.e., either test positive) the most sensitive and least specific; sensitivity is lower than cytology alone (≥ASCUS), potentially leading to unignorably large numbers of CIN2+ women undiagnosed (false negatives, FNs) depending on the prevalence of CIN2+ in a screened population [17,24].

Conversely, the AND rule (i.e., both test positive), the least sensitive and most specific, would increase the number of healthy women misclassified as CIN2+ (false positive, FPs) while identifying only a few more women with CIN2+ [24,25,26].

Shortcoming of such FPs results include unnecessary colposcopy, further triage tests, and psychological burden, unjustified in low-risk setting populations. Combination algorithms of (i). cytology ≥ LSIL or [ASCUS cytology with hrHPV positive test] or (ii). hrHPV 16/18 or [HPV other than 16/18 and cytology ≥ ASCUS] are found equal/no less specific than hrHPV testing alone and more sensitive than cytology alone [26,27].

Current European guidelines for quality assurance in CC screening (2015) recommend avoidance of co-testing (HPV and cytology primary testing) at any given age. Women testing primary HPV positive should be tested for cervical cytology (cytology triage) then, according to the result, subsequent referral to repeat testing or colposcopy [28,29,30]. Women with pre-invasive or more severe cytology at triage (ASC-H, HSIL, AIS) should be referred to colposcopy. Women with minor cytological abnormalities at triage (ASC.US, AGC, LSIL) should be retested at 6–12 months or referred to colposcopy.

Protocols using either HPV testing or cytology alone in repeat testing should be completed with colposcopy in case of an HPV positive test or cytology ≥ ASCUS, respectively. Self-sampling in screening programs using HPV primary testing displays sufficient clinical accuracy with comparable sensitivities to physician-collected samples, albeit a slightly lower specificity, particularly important for poor compliers and women in rural areas with limited access to health centers [28,29,30].

## 2. Materials and Methods

An electronic database search using PubMed, Medline, and Embase was conducted up to July 2023. A search algorithm was developed incorporating the following terms: “Cervical Cancer”; “Surgery treatment”; “Sentinel lymph node mapping”; “Early-stage”; “Fertility preservation”; “HPV”; “Pre-cancerous lesion”; “HPV Vaccination”; “Cervical intraepithelial lesion”. All pertinent articles evaluating the diagnostic and therapeutic approaches centering on cervical cancer have been included in this review (Figure 1). All original studies, meta-analyses, systematic reviews, and case reports published in English were considered. The analysis considers evaluating the different surgical strategies available for CC and their association with the best results. The reference lists were systematically reviewed to identify other studies for potential inclusion in this narrative review. We adhered to the quality standards for narrative reviews, as defined and quantified by “SANRA—a scale for the quality assessment of narrative review articles” [31]. A total of 19 studies has been included in this narrative overview (Figure 1).

## 3. Results

### 3.1. Excision Procedure

According to the 2019 American Society for Colposcopy and Cervical Pathology (ASCCP) management consensus guidelines, pivot criteria for clinical decision-making in case of abnormal CC screening tests is CIN3+ risk. CIN3+ encompasses CIN3, adenocarcinoma in situ, and CC. On the other hand, most CIN2 lesions, especially in young women (<30 years old), regress spontaneously [9].

The Indications for a colposcopy to be performed are risk-based. The choice of performing colposcopy is related to the underlying risks for precancerous cervical lesions based on their cytological results, the HPV testing if it was performed, and personal history of cervical dysplasia. Colposcopy allows us to diagnose if dysplasia is present and its severity. Indications for colposcopy include the following: evaluation of women with an abnormal Pap test and in case of positivity for high-risk HPV DNA; evaluation of a suspicious-appearing cervix and postcoital/postmenopausal bleeding, even if the Pap smear is normal; unexplained abnormal lower genital tract bleeding; persistent inflammatory/unsatisfactory cervical cytology despite appropriate treatment; identification and management of subclinical papillomavirus infection; conservative management of intraepithelial neoplasia; dentification and management of a vaginal lesion in case of concomitant cervical neoplasia; post-treatment follow-up [30,31,32,33].

Expedited treatment is preferred for HPV16-positive HSIL cytology or HPV-positive HSIL cytology, regardless of genotype, if never or rarely screened.

Women with minor screening abnormalities indicating HPV infection with low risk of underlying CIN3+ should repeat testing or co-testing at 1 year; for example, HPV positive, LSIL with previous negative HPV test or co-test [30].

As evident, in the case of abnormal CC screening test, a shift from results-based management to risk-based management is encouraged and a tailored approach, including current and past screening test results and medical history, should be applied. Table 1 reports all diagnostic and therapeutic strategies in case of cervical lesions evaluated by colposcopy, all of which can be performed during the colposcopic examination.

### 3.2. Trachelectomy and SLN Mapping

In young patients (40% are under 45 years old), the reference treatment to spare fertility and at the same time reduce the risk of CC is represented by trachelectomy (vaginal, abdominal, or minimally invasive approach). This treatment consists of the removal of the entire cervix, parametric tissue, and vaginal cuff—preserving the uterus, ovaries and tubes—with SLN mapping with or without bilateral pelvic lymphadenectomy [23]. The selection criteria for this type of surgery are young age (less than 40 years old), fertile patients with a reproductive desire, stage IA1 with LVSI or IA2, IB1 stages (tumor size < 2 cm), and MRI not showing parametrial invasion or metastases to lymph nodes or other sites. The debate is still open and the need for further trials is truly felt; in fact, in the literature, radical trachelectomy (RT) and radical hysterectomy (RH) for tumors up to 4 cm showed similar recurrence-free survival and disease-specific survival, and with the functional results too, such as disease-free survival, overall survival, and recurrence rate [23,24,25,26,27,28]. In the paper of Feng et al. [34], the recurrence rates and mortality rates of RT were similar to that of RH; patients with tumors 2–4 cm in diameter were more likely to receive RH, but, moreover, this treatment was significantly associated with the likelihoodof intraoperative blood transfusion. Furthermore, for lesions 2–4 cm and a desire for fertility, RAT (radical abdominal trachelectomy) can be a feasible alternative to RH under fully informed consent. The possibility is also reported for early-stage patients with CC whose lesions are less than 2 cm to receive RVT (radical vaginal trachelectomy) [29,30,31,32,33,34], which, according to some authors, is associated with better reproductive outcomes. Among the data with the greatest impact on pregnancy rates after conservative treatment are those of Shinkai et al. on RT for 71 patients and 28 pregnancies [35]. All the RTV were performed safely without serious complications. The median time to be pregnant after RT was 29.5 months. Thirteen patients (46%) became pregnant without artificial insemination or assisted reproductive technology. Cesarean section was performed for all of them. The median time of pregnancy was 34 weeks, and emergent cesarean section was performed in seven patients (25%). Moreover, the median birth weight was 2156 g. Four patients had trouble with cervical cerclage, and they had sudden premature preterm rupture of the membrane (pPROM) during the second trimester of pregnancy. They underwent transabdominal cerclage (TAC) for all of them with careful management for the prevention of uterine infection. One patient had a recurrence of cancer during pregnancy [35].

### 3.3. Laparotomy Hysterectomy versus Laparoscopy Management

In 1912, Wertheim published for the first time the description of the treatment of radical hysterectomy. Later, in 1944, Meigs described bilateral pelvic lymphadenectomy with the removal of four major ureteral, hypogastric, obturator, and iliac groups. Thus, radical hysterectomy was called Wertheim-Meigs, which includes the removal of uterus, vagina, and parametria. Since then, laparotomy radical hysterectomy has been the reference approach for CC treatment for decades [28,29] (Table 2).

In recent times, through the analysis of retrospective studies, laparoscopic and robotic approaches to radical hysterectomy have become common and guidelines have been integrated, starting as options for the treatment of IA2-IIA CC by the National Comprehensive Cancer Network (NCCN) and the European Society of Gynecological Oncology (ESGO) [34,35,36]. The most important retrospective data showed less blood loss, less hospitalization, and fewer post-operative complications, and no worsening of overall survival (OS) [37,38,39,40] compared to open surgery.

However, things changed when the data were taken into account in a prospective approach, with the “LACC trial” (Laparoscopic Approach to Cervical Cancer trial) [41].

The LACC trial, in fact, is a prospective study of randomized women with stage IA1 (lymphovascular invasion), IA2, or IB1 CC and a histologic subtype of squamous cell carcinoma, adenocarcinoma, or adenosquamous carcinoma, that compares minimally invasive surgery (MIS) and open surgery. About 600 patients were studied in total and 91% had stage IB1 disease and most MIS patients were treated laparoscopically (only 15% robotic-assisted).The two groups had similar characteristics for cancer and the use of adjuvant therapy. The disease-free survival rate at 4.5 years was significantly worse in the MIS group (86% vs. 96.5%). MIS approach was associated with a hazard ratio (HR) of disease recurrence or death of 3.74 [95% confidence interval (CI): 1.63–8.58], and this remained after adjusting for potentially confounding factors; OS at 3 years was lower in the MIS group (93.8% vs. 99.0%), and after this important information, many centers have tried to clarify by adding other data [41,42,43,44,45,46].

In conjunction with the “LACC trial” results, another turning point was given by the work of Melamed et al. [47]. In fact, they published an epidemiologic study evaluating the association of survival rates and a minimally invasive approach for radical hysterectomy in the same journal. This was a cohort study of women who underwent radical hysterectomy for IA2 or IB1 CC from 2010 to 2013 at accredited hospitals and an interrupted time series analysis of women undergoing radical hysterectomy from 2000 to 2010 using the SEER database (The Surveillance, Epidemiology, and End Results (SEER) Program). In this analysis, a four-year relative survival rate for women undergoing radical hysterectomy was stable, but after adoption (after 2006) of MIS, it declined by a rate of 0.8% per year (*p* = 0.01). This corroborative study suggested that MIS radical hysterectomy was associated with a shorter OS. They found a significant increase in 4-year mortality in women undergoing MIS radical hysterectomy compared with open surgery (9.1% vs. 5.3%, HR: 1.65, 95% CI: 1.22–2.22) [47].

### 3.4. Laparotomy Hysterectomy versus Robotic Management

Falconer et al. Compared a robotic-assisted laparoscopic treatment group to one treated with open surgery. In this analysis, they have included women over 18 years old diagnosed with Stages IB1, IB2, and IIA1 CC according to FIGO 2018 [42]. In this important study, different outcomes were examined, including oncological safety for robot-assisted surgery vs. standard laparotomy. The Authors found a recurrence-free survival of about 85%, but the analysis is still ongoing, and the sample (127) to be reached does not yet have significant statistical power. For this reason, the scientific community is waiting for further results [42].

One of the question is: if robotic surgery is compared with laparoscopic approach, do different results emerge in oncological terms? Chen et al. [43] compared clinical efficacy and safety of robotic radical hysterectomy (RRH) in 216 patients with laparoscopic radical hysterectomy (LRH) in 342 patients in terms of complications with 20 (9.65%) patients in the first group and with 60 (17.59%) patients in the second one, and in terms of recurrence rates, with 15.7% for RRH and 12% for LRH. Furthermore, the oncological outcomes were evaluated with non-statistically significant results [*p* = 0.407 for RRH and *p* = 0.28 for LRH], with PFS of 28.91 ± 15.68 months in the first group and 28.34 ± 15.13 months in the second one and OS of 92.13% in RHH and 94.45% in LRH groups, respectively.

In conclusion, RRH seems to be associated with significantly less operative time and blood loss than LRH, but these types of surgery have similar complication rates, OS, and PFS [43]. Further studies and insights will better clarify the advantages of robotic surgery for the treatment of this type of tumor in the future.

### 3.5. Uterine Manipulator for Early-Stage Cervical Cancer

Liu et al. reported in a recently meta-analysis a lower recurrence-free survival (RFS) compared to open radical hysterectomy when this was performed in a minimally invasive manner, with the rating of six observational studies and 2150 women.

It is estimated that the minimally invasive surgery group had a significantly higher risk of cancer relapse compared with the open surgery group (HR 1.55, 95% CI 1.15–2.10) [44].

Kampers et al. evaluted about 30 studies, including both prospective studies and randomized-control trials on open surgery (AH), laparoscopic surgery (LH), and robotic surgery (RH) [45].

Moreover, further subgroups were also selected, including an LH high-risk group with use of a manipulator, an LH intermediate-risk group without use of a manipulator but with intracorporal colpotomy, and an LH low-risk without use of a manipulator but with vaginal colpotomy. In conclusion, for this subclassification, there was an OS significantly higher in LH low-risk compared to higher risk groups.

Therefore, OS rates were comparable in AH and LH low-risk groups [45,46,47,48]. In the AH group, DFS was higher, compared to the LH group, and so the use of protective measures was associated with increased DFS in laparoscopy [45], as reported in the literature [49,50,51,52,53,54].

It has also been taken into consideration by several authors [55,56], such as D’Asta et al. [57], who carried out a retrospective study on patients surgically treated for early CC from 2014 to 2017. In this study, the patients presented squamous or adenosquamous CC, FIGO stage from Ia1 to Ib2, size < 4 cm, ECOG status 0–1. Thirty-one patients were enrolled and divided into two groups, three with and 28 without an intra-uterine manipulator. Among them, 12 women had cancer in situ (IA1), 19 had early stage CC, including two cases of stage IA2, 10 of stage IB1, and seven of stage IB2, according to the FIGO classification. Three cases of recurrence were described, none of them with uterine manipulator. After 5 years of follow-up, the recurrence rate in patients with minimally invasive surgery was 10%, but the use of a uterine manipulator was not related to a higher level of recurrence rates.

DFS and OS in laparoscopy appear to depend on the use of protective systems and the technique used.

In this debate, Wang et al. [58] investigated whether the use of a uterine manipulator (UM) or intracorporeal colpotomy conferred inferior short-term survival among patients treated for early-stage CC. They reported a retrospective cohort study with 1169 patients (stage IB1 to IB2 CC), and all these patients underwent minimally invasive radical hysterectomy and pelvic lymphadenectomy.

These 1169 patients diagnosed with preoperative stage IB1 to IB2 CC were primarily treated with surgery between 2018 and 2019. The eligible patients had a median age of 48 years, and the median follow-up time was 34 months. The 2-year overall survival rate of the patients with pathologic stage IB1 and IB2 was 99.8% and 98.8%, respectively, according to the 2018 International Federation of Gynecology and Obstetrics staging system. Univariable analysis showed that the UM group had a 7.6-times higher risk of death than that of the manipulator-free group (*p* = 0.006), but multivariable analysis revealed that only tumor size and parametrial involvement were independent risk factors for overall survival. Lastly, short-term survival outcomes in women undergoing minimally invasive radical hysterectomy for the treatment of early-stage CC did not differ when a UM was avoided or when a protective colpotomy was performed. There was no statistically significant difference in survival between patients who underwent intracorporeal and protective colpotomy [58].

### 3.6. Fertility Sparing-Surgery

In young patients, the trachelectomy treatment (vaginal, abdominal approach, or minimally invasive approach) consists of the surgical treatment on the cervix, vaginal cuff, and parameters—preserving the uterus, ovaries, and tubes—with SLN mapping with or without bilateral pelvic lymphadenectomy [31,59,60,61].

The selection criteria for this type of surgery includes: less than 40 years old, fertile patient with a reproductive desire, and tumor size < 2 cm and with negative MRI regarding the expression of tumor cells on parameters, lymph nodes, or distant tissues [61,62].

In the literature, as mentioned above, there are conflicting results, with similar RFS and disease-specific survival between RT and RH (tumors up to 2–4 cm), and the same functional results (disease-free survival, overall survival, and recurrence rate) were obtained if one compared the open approach with the MIS [63,64,65] (Table 3).

Additionally, with a total of 33 patients, He et al. reported that there are no statistically significant difference in oncological outcome between the laparoscopy and abdominal surgery. LRT resulted in less blood loss and a decreased length of hospital stay. Furthermore, the clinical pregnancy rate in the ART group was significant higher than that in the LRT group [62].

According to the opinion of many scientists, infiltration of less than half of the cervical stroma is the limit for a safe trachelectomy, because it is necessary to have a 1 cm free margin; some authors suggest margins of only 5–8 mm may be sufficient, but this is still debatable [59]. All types of trachelectomy should save a good proportion of healthy stroma because the possibility of successful pregnancy is higher. In fact, the preservation of cervical stroma is fundamental to a lower risk of cervical incompetence, premature delivery, premature rupture of membranes, and other pathological conditions of an infectious type [59,60].

In the literature, parametrial involvement in IB1 tumors ranges from 6 to 13%, and we reiterate to conclude that factors which potentially correlate with parametrial tumor spread at the time of radical hysterectomy include lymph node status, size of tumor, deep stromal invasion, lymph vascular space invasion, grade, stage, histology, and presence of residual tumor [59,60,61].

Patients with CC that has spread to the parametria require adjuvant chemoradiation and therefore lose the benefit of the “fertility-sparing” aspect of the surgery. In these patients, there may be an increased risk of complications. Unfortunately, most of the characteristics that increase the risk of spread, such as stromal and vascular invasion, may not be reliably determined preoperatively. For this reason, the use of conservative surgery for the treatment of cervical lesions remains one of the most modern and active debates in the gynecological oncological literature [59,60].

## 4. Discussion

The role of minimally invasive surgery (MIS) has changed after the results of the LACC Trial published in 2018 [39,40,41,42,43]. It has strongly influenced the debate about management of CC, concluding that minimally invasive surgery (MIS) was associated with a higher rate of recurrence and a higher rate of all-cause death with a four-fold higher rate of recurrence and with a six-fold higher rate of all-cause death when compared to open approach [66,67].

The statistical power of the population enrolled in a trial that has so deeply changed the approach to these tumors should be enough to place a statistically significant condition on each aspect, including a patient with low-risk profile, characterized by tumor size < 2 cm, no LVSI, and invasion depth < 10 mm. The “LACC trial” was not statistically powered for patients entering this subgroup [44,45,46]. For this reason, many experts believe that the debate remains open for the use of MIS, and, in some papers, a tumor size of 20 mm represents the only selection factor for surgical treatment [47,48].

Another point of criticism of the LACC trial is that even surgeons without great experience of minimally invasive surgery performed the operation, with subsequent weight and/or bias on results.

We deepen this discourse by recalling the criteria for participation, which were the submission of 10 cases of Total Laparoscopic Radical Hysterectomy and Total Robotic Radical Hysterectomy (TLRH/TRRH) and a total of two unedited videos of (TLRH/TRRH) to be sent for the attention of the management committee.

The patients diagnosed with squamous, adenocarcinoma, or adenosquamous CC Stage IA1 LVSI, IA2, IB1 were randomized in total abdominal radical hysterectomy (312) and total laparoscopic/robotic radical hysterectomy (319) to compare the surgical and oncological outcomes [49,50].

Liu et al. divided patients who underwent laparoscopy into uterine manipulator and uterine manipulator-free groups, with an incidence of LVSI in 45.22% and 48.35%, respectively. The surgical approach and use of a uterine manipulator are not associated with LVSI in surgery for early-stage CC [44].

There are several theories that favor open surgery over MIS; among these, that the routine use of a uterine manipulator might increase the propensity for tumor spillage. In addition, an effect of the insufflation gas (CO_2_) on tumor cell growth or spread has been suggested in previous studies, and other theories are associated with the time of colpotomy [51,52,53,54,55].

Casarin et al. [68] reported that, in women with early-stage CC, a predictor of recurrence after LRH is the presence of high-volume disease, while a protective role is played by preoperative conization and the absence of residual disease.

However, there is agreement among experts regarding the need for further results to define safety methods which are able to reduce and/or prevent the dispersion of neoplastic cells during procedures.

In this debate, the “SUCCOR study”, an international European cohort observational study comparing minimally invasive surgery and open abdominal radical hysterectomy in patients with stage IB1 CC, reaffirms that MIS in CC increases the risk of relapse and death versus open surgery. However, similar results between MIS and open surgery have been obtained by not using the uterine manipulator and with safety maneuvers during the colpotomy to avoid the spread of the tumor [69].

Regarding MIS in CC treatment, there is the possibility of injecting a dye (blue, indocyanine green, or radiopharmaceutical substances such as technetium 99) into the cervix [70] in early-stage CC according to the following selection criteria: tumors < 4 cm, no suspicious lymph nodes on imaging, bilateral evidence of SNL, and availability of ultrastaging [71] (Table 4).

If the pathological examination of the frozen section reveals positive findings [72], the patient must undergo radio-chemotherapy, and surgical treatment cannot be performed [73,74].

According to data available in the literature [65], the prevalence of SLN metastases at an early stage is 21%. In fact, the role of the sentinel lymph node emerges with excellent detection and sensitivity rates (94%), and a negative predictive value of 91–100%. The ongoing trials that are trying to give a complete answer about the situation are “Sentix”, “Phenix”, and “Senticol III”. Sentinel lymph node mapping is feasible and its role will become increasingly predominant in the future, offering an accurate anatomical staging and identifying potential metastatic lymph nodes outside the normal areas of lymphadenectomy.

In early CC, OS rate is 70–90%, but for prognosis, the staging, tumor size, lymph node involvement, depth of stromal invasion, and LVSI [75] are essential factors to take into account [76].

In addition, size is also an important prognostic factor [77], as it is related to parametrium and lymph node involvement and decreasing survival rates. Lymph node involvement is considered another important factor in terms of prognosis, as the five-year survival rate for patients with early-stage CC without lymph node involvement is about 90% [78,79].

## 5. Conclusions

Progression from HPV infection to cancer normally takes 15–20 years. This long natural history with a prolonged precancerous phase permits early detection and treatment through population screening.

CC is largely preventable through local treatment of screen-detected cervical pre-invasive lesions (high-grade cervical intraepithelial neoplasia, HG CIN).

Implementation of screening strategies and distribution of prophylactic HPV programs are major drivers of a reduced epidemiological burden of HPV infections and related neoplasms, at least in high-income countries.

When prevention fails, the treatment of CC, especially in the early stages, is at the center of the scientific debate, and the best therapeutic choice can often be surgery. Further insights are emerging as described on the role of parametrectomy in patients with tumors up to 2 cm, and with respect to the role of SLN mapping.

In selected young patients, conservative treatment in case of a desire for offspring remains a fundamental factor to be taken into consideration.

The best surgical treatment, when needed according to disease stage, is controversial, although, to date, open surgery is preferred to MIS and, within this latter, there is no major difference between laparoscopy and robotic surgery.

Patients from high-income countries might have access to the best possible treatments according to the highest level of evidence, compared to patients from low-income countries. For this reason, in low- or middle-income countries it is essential to increase the level of prevention by raising vaccination and screening test coverage for HPV infections and cancer in the population.

## Figures and Tables

**Figure 1 healthcare-11-02942-f001:**
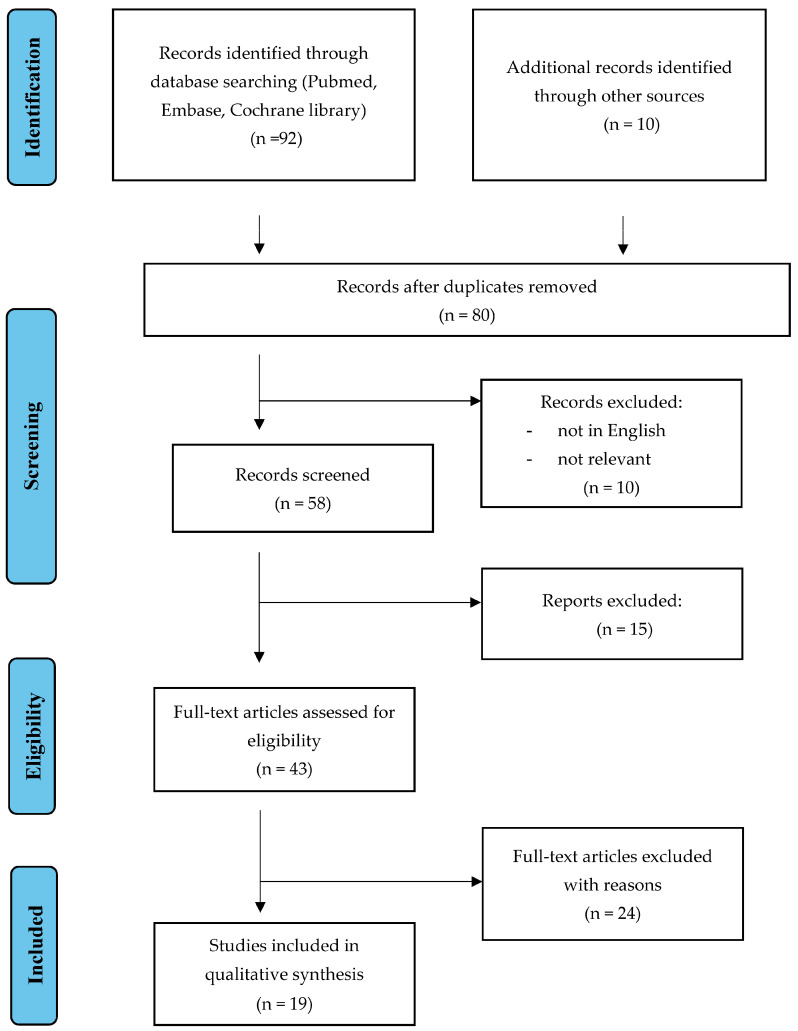
Flow diagram of review search.

**Table 1 healthcare-11-02942-t001:** Local surgical diagnostic and therapeutic procedures on the cervix.

*Procedure Type*	Description
** *Punch biopsy* **	Surgical procedure consisting in the removal of a round-shaped tissue sample for pathological analysis.
** *Endocervical* ** ** *curettage* **	Surgical procedure consisting in the collection of tissue from the cervical canal to find a glandular lesion or an endocervical squamouslesion which cannot be found with abiopsyincolposcopy.
** *Loop electrosurgical excision procedure (LEEP)* **	Diagnostic and therapeutic technique employing energy for removing atypical cells from the cervix for subsequent histological examination.
** *Cone biopsy* **	Diagnostic and therapeutic technique in which a cone-shaped piece of tissue from the cervix and cervical canal is removed. The aim of this procedure is the removal of precancerous lesion or early-stage cancer. A synonym for cone biopsy is cervical conization.

**Table 2 healthcare-11-02942-t002:** Types of major surgery in the treatment ofCervical Cancer.

*Procedure Type*	Description
** *Total hysterectomy* ** ** *(TH)* **	Removal of the uterus and the cervix. Further subclassification according to access techniques in: 1. Total vaginal hysterectomy 2. Total abdominal hysterectomy 3. Total laparoscopic hysterectomy
** *Radical hysterectomy* ** ** *(RH)* **	Removal of the uterus, cervix, part of the vagina, and a wide area of ligaments and tissues around them; also ovaries, fallopian tubes, nearby lymph nodes.
** *Modified radical* ** ** *hysterectomy (MRH)* **	Removal of the uterus, cervix, upper part of the vagina, and ligaments and tissues that closely surround these organs; also ovaries, fallopian tubes, nearby lymph nodes.
** *Radical trachelectomy* ** ** *(RT)* **	Removal of the cervix, nearby tissue, the upper part of the vagina with/without removal of regional lymph nodes.

**Table 3 healthcare-11-02942-t003:** Types of treatment according to cervical cancer stage.

Cervical Cancer Stage	Treatment Description
**Treatment of stage IA cervical cancer**	**STAGE IA1** Cold knife conization: fertility-sparing procedure Total hysterectomy with or without bilateral salpingo-oophorectomy **STAGE IA2** Modified radical hysterectomy and removal of lymph nodes Radical trachelectomy: fertility-sparing surgery and removal of lymph nodes Internal radiation therapy: patients who cannot have surgery for different reasons
**Treatment of stages IB and IIA cervical cancer**	Radiation * therapy at the same time as chemotherapy Radical hysterectomy and removal of pelvic lymph nodes with or without radiation therapy to the pelvis with chemotherapy ** Radical trachelectomy: fertility-sparing surgery Radiation therapy alone * *external radiation therapy and/or combination of external-internal radiation therapy.* *** chemotherapy drugs: cisplatin or carboplatin and/or radiation therapy.*
**Treatment of stages IIB, III, and IVA cervical cancer**	Radiation therapy given at the same time as chemotherapy Surgery to remove pelvic lymph nodes followed by radiation therapy with or not chemotherapy
**Treatment of stage IVB cervical cancer**	Radiation palliative therapy Chemotherapy *** and the targeted therapy drug bevacizumab (palliative therapy) **** cisplatin, carboplatin, ifosfamide, irinotecan, gemcitabine, paclitaxel, and topotecan.*
**Treatment of recurrent cervical cancer**	Immunotherapy drug pembrolizumab Radiation therapy +/− chemotherapy **** Pelvic exenterationfor who cannot have radiation therapy ***** cisplatin, carboplatin, ifosfamide, irinotecan, gemcitabine, paclitaxel, topotecan, and vinorelbine.*

**Table 4 healthcare-11-02942-t004:** The *“Copernican revolution”* on the surgical treatment of Cervical Cancer: five works that have impacted the way of thinking on this topic.

Study/Year	Country	Type of Study	Stage/Types of Tumors	Sample Size, n°	Age (Years-Mean ± SD or Median (Range))	Surgical Treatment	Primary Outcomes	Results
Ramirez et al., 2018 [41]	US	Prospective (randomized trial) *“LAAC TRIAL”*	*Stage IA1, IA2,* or *IB1 CC* and a histologic subtype of squamous cell carcinoma, adenocarcinoma, or adenosquamous carcinoma	319 (MIS) versus 312 (AH)	46.0 ± 10.6 46.1 ± 11.0	MIS versus AH	The rate of DFS and OS	MIS reported lower rates of DFS and OS
Melamed et al., 2018 [47]	US	Retrospective (cohort study)	*Stage IA2* or *IB1CC* and a histologic subtype of squamous cell carcinoma, adenocarcinoma, or adenosquamous carcinoma	2461 patients	NA	MIS versus AH	The rate of OS	MIS reported a lower rates of OS
Falconer et al., 2019 [42]	Sweden	Prospective multi- institutional international open-label randomized clinical trial *“RAAC TRIAL”*	*Stages* *IB1, IB2, IIA CC* and a histologic subtype of squamous, adenocarcinoma, or adenosquamous	*NA* *(interim analysis):* 3 years after the first patient is randomized or when 300 patients have been included in the study	NA	TLRH versus AH	Recurrence-free survival at 5 years	The clinical non-inferiority margin is defined with a cut-off by >7.5%
Casarin et al., 2020 [68]	Italy	Retrospective multi-institutional study	*Stage IA1, IA2* and *IB1 CC,* and a histologic subtype of squamous, adenocarcinoma, or adenosquamous	428 patients	45 (recurence NO) - 48 (recurence YES)	TLRH	To assess predictors of recurrence following TLRH	Independent predictor of recurrence after TLRH (high-volume disease)
Chiva et al., 2020 [69]	Europe *(multicenter)*	European multicenter retrospective observational cohort study *“SUCCOR study”*	*Stage IB1 CC* and a histologic subtype of squamous, adenocarcinoma, or adenosquamous	693 patients	48.3 (23–83)	MIS versus AH	To compare DFS, OS, and the role of uterine manipulator	MIS in CC increased the risk of relapse and death compared with AH

[MIS: Minimally invasive surgery; AH: Open surgery; CC: Cervical Cancer; LPS: laparoscopic surgery; LHR: laparoscopic radical hysterectomy; TLRH: Total Laparoscopic Radical Hysterectomy; TRRH: Total Robotic Radical Hysterectomy; DFS: disease free survival; OS: overall survival].

## Data Availability

The present review was based on published articles. All summary data generated during this study are included in this published article. Raw data used for the analyses are available presented in the original reviewed articles.

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
