# Peer review of "Surgical Treatment for Early Cervical Cancer in the HPV Era: State of the Art"

_healthcare, 2023, doi:10.3390/healthcare11222942_

Round 1

Reviewer 1 Report

Reviewer’s Comment

1.       Line no.18 and 19 are repeated in the abstract

2.       Line no. 33. The term screening may be removed from Keywords

3.       The authors have initially screened 62 articles, finally landed with 17 articles only , 72 reference quoted in this review. Are  these 17 articles  enough to conclude the findings?

4.       Line no.134 to 137 has 7 references (8-14) but some of them are mis matching. ASCCP 2019 citation is enough.

5.       Line no 139-142 has 9 references, kindly verify

6.       Table 1, Table 2 & Table 3 Kindly add heading in the each column head.

7.       In Discussion section (line No. 378-380) kindly rewrite the statement. It is not conveying the right meaning

8.       Line no. 390, what are TLRH/TRRH?

9.       The discussion & conclusion could have been written better.

Author Response

September 25, 2023

Dear Editors,

We received the reviewer’s comments regarding our manuscript submitted for consideration for publication entitled "Surgical treatment for Early Cervical Cancer in the HPV era: state of the artand we are grateful for the opportunity to revise our work. We would like to thank the reviewers for considering the work interesting and for taking the time to make those very appropriate comments to improve it. We have modified the text of the manuscript to address all the areas identified by the reviewers. We believe these revisions have adequately addressed the raised points, and we hope that the revised version of the manuscript is now considered acceptable for publication.

ACCORDING TO REVIEWER 1’S SUGGESTIONS:

  1. “Line no.18 and 19 are repeated in the abstract”
  2. We eliminated this repetition in the abstract.
  3. Line no. 33. The term screening may be removed from Keywords”
  4. We eliminated this Keyword.
  5. The authors have initially screened 62 articles, finally landed with 17 articles only , 72 reference quoted in this review. Are these 17 articles  enough to conclude the findings?”
  6. We believe that this consideration is correct (in fact we have further integrated the works in the bibliography), but the number of works analyzed is sufficient to focus attention on the surgical approach method. Let's think about the fact that the debate on minimally invasive surgery among the therapeutic options has only recently opened, and so, the number of works is in keeping with the times. Moreover, after a careful integration with other important material of the international literature we have added two other works to the analysis (Casarin et al. / Chiva et al.). Thank you for allowing other insights.
  7. Line no. 134 to 137 has 7 references (8-14) but some of them are mis matching. ASCCP 2019 citation is enough.”
  8. As you asked, we moved reference 25 (ASCCP) to line no. 134-137. We also arranged the bibliography as you pointed out.
  9. Line no 139-142 has 9 references, kindly verify”
  10. 5. We have corrected according to your requests.
  11. Table 1, Table 2 & Table 3 Kindly add heading in the each column head”.
  12. Ok. You can see “heading in the each column head” in the tables as reported.
  13. In Discussion section (line No. 378-380) kindly rewrite the statement. It is not conveying the right meaning.
  14. I proceeded to rewrite as requested.
  15. Line no. 390, what are TLRH/TRRH?
  16. Total Laparoscopic Radical Hysterectomy and Total Robotic Radical Hysterectomy: I specified in the text
  17. The discussion & conclusion could have been written better.
  18. As recommended we did a revision of the text, with careful and appropriate additions and revisions, we hope that now the work is ready for publication.

******

We agree with the reviewer. All the points highlighted by the reviewer represent the future aims of our research line. Our study represented a promising starting point and showed that this approach could discriminate between patients affected by endometriosis and not.  However, our study shows several weak points, including those highlighted by the reviewer.

We elaborated the conclusion section describing strengths e weakness points of the paper including those mentioned by the reviewer.

Once again, we would like to thank the Reviewers for the precious suggestions and the Editor for allowing us to improve our manuscript. We hope our paper now has the quality to be accepted for publication in your prestigious Journal.

We remain at your disposal for any further detail you might consider essential to clarify.

On behalf of the co-Authors,                                                                   

Luigi Della Corte*

*[email protected]; [email protected] (Luigi Della Corte, M.D. 1Department of Neuroscience, Reproductive Sciences and Dentistry, School of Medicine, University of Naples “Federico II”, 80131 Naples, Italy. ORCID: 0000-0002-0584-2181.)

Reviewer 2 Report

Dear authors,

your paper is interesting, but :

- in the abstract the first sentance is repeted

- the introduction part is too short and should contain more information about surgery and less HP , so I sugest to modify all the introduction part

- the article is welll written but  there is less novelty

about this article 

- the discussion part is short and should include more data from other studies

English is good

Author Response

September 25, 2023

Dear Editors,

We received the reviewer’s comments regarding our manuscript submitted for consideration for publication entitled "Surgical treatment for Early Cervical Cancer in the HPV era: state of the artand we are grateful for the opportunity to revise our work. We would like to thank the reviewers for considering the work interesting and for taking the time to make those very appropriate comments to improve it. We have modified the text of the manuscript to address all the areas identified by the reviewers. We believe these revisions have adequately addressed the raised points, and we hope that the revised version of the manuscript is now considered acceptable for publication.

ACCORDING TO REVIEWER 2’S SUGGESTIONS:

  1. “In the abstract the first sentance is repeted.”
  2. We eliminated this repetition in the abstract.
  3. “The introduction part is too short and should contain more information about surgery and less HP, so I sugest to modify all the introduction part.”
  4. As you requested we have modified introduction, in order to maintain a clinical approach, to give more attention to the role of HPV we divided the introduction into two sub-paragraphs for deepening the role of screening tests.
  5. “The article is well written but there is less novelty about this article.”
  6. We realise that this work does not seem to introduce anything new, but very often many colleagues find it difficult to find the right information on the treatment of this disease. The creation of such a topic allows an easier understanding of such a complex topic. We hope that the data insights are to your liking.
  7. 4. “The discussion part is short and should include more data from other studies.”
  8. As requested we have made an integration into the text. After a careful revision of the text, we say that this is now ready for publication.

******

We agree with the reviewer. All the points highlighted by the reviewer represent the future aims of our research line. Our study represented a promising starting point and showed that this approach could discriminate between patients affected by endometriosis and not.  However, our study shows several weak points, including those highlighted by the reviewer.

We elaborated the conclusion section describing strengths e weakness points of the paper including those mentioned by the reviewer.

Once again, we would like to thank the Reviewers for the precious suggestions and the Editor for allowing us to improve our manuscript. We hope our paper now has the quality to be accepted for publication in your prestigious Journal.

We remain at your disposal for any further detail you might consider essential to clarify.

On behalf of the co-Authors,                                                                    

Luigi Della Corte*

*[email protected]; [email protected] (Luigi Della Corte, M.D. 1Department of Neuroscience, Reproductive Sciences and Dentistry, School of Medicine, University of Naples “Federico II”, 80131 Naples, Italy. ORCID: 0000-0002-0584-2181.)

Reviewer 3 Report

The authors presented a study to investigate the screening, diagnosis and surgical approach of cervical cancer.
In my honest opinion, the topic is interesting enough to attract the readers’ attention and it is of the utmost importance given the efforts made by the WHO for the prevention and early diagnosis of cervical cancer and precancerous cervical lesions.
References are relevant to the research, however, I suggest adding some insight about the use/avoidance of minimally invasive surgery for cervical cancer, considering the available pieces of evidence and the histotype of the disease (see PMID: 36293758).
The methodology is accurate and the conclusions are consistent with the arguments presented. The tables and figures are clear and interesting.
References are appropriate.
I suggest another round of language revision, in order to correct few typos and improve readability.

Minor editing of the English language is required to make the work clearer and more readable.

Author Response

September 25, 2023

Dear Editors,

We received the reviewer’s comments regarding our manuscript submitted for consideration for publication entitled "Surgical treatment for Early Cervical Cancer in the HPV era: state of the artand we are grateful for the opportunity to revise our work. We would like to thank the reviewers for considering the work interesting and for taking the time to make those very appropriate comments to improve it. We have modified the text of the manuscript to address all the areas identified by the reviewers. We believe these revisions have adequately addressed the raised points, and we hope that the revised version of the manuscript is now considered acceptable for publication.

ACCORDING TO REVIEWER 3’S SUGGESTIONS:

“The authors presented a study to investigate the screening, diagnosis and surgical approach of cervical cancer.
In my honest opinion, the topic is interesting enough to attract the readers’ attention and it is of the utmost importance given the efforts made by the WHO for the prevention and early diagnosis of cervical cancer and precancerous cervical lesions”.

1.“References are relevant to the research, however, I suggest adding some insight about the use/avoidance of minimally invasive surgery for cervical cancer, considering the available pieces of evidence and the histotype of the disease (see PMID: 36293758)”.

  1. We integrated this work as suggested.

2.”The methodology is accurate and the conclusions are consistent with the arguments presented”.

“The tables and figures are clear and interesting”.

“References are appropriate”.
“I suggest another round of language revision, in order to correct few typos and improve readability”.

  1. We can only thank you for these comments and we provide a revision of the language.

******

We agree with the reviewer. All the points highlighted by the reviewer represent the future aims of our research line. Our study represented a promising starting point and showed that this approach could discriminate between patients affected by endometriosis and not.  However, our study shows several weak points, including those highlighted by the reviewer.

We elaborated the conclusion section describing strengths e weakness points of the paper including those mentioned by the reviewer.

Once again, we would like to thank the Reviewers for the precious suggestions and the Editor for allowing us to improve our manuscript. We hope our paper now has the quality to be accepted for publication in your prestigious Journal.

We remain at your disposal for any further detail you might consider essential to clarify.

On behalf of the co-Authors,                                                                    

Luigi Della Corte*

*[email protected]; [email protected] (Luigi Della Corte, M.D. 1Department of Neuroscience, Reproductive Sciences and Dentistry, School of Medicine, University of Naples “Federico II”, 80131 Naples, Italy. ORCID: 0000-0002-0584-2181.)

Reviewer 4 Report

Surgical treatment for Early Cervical Cancer in the HPV era: state of the art

Reviewer's report:

The study emphasizes concepts significant to oncology gynecologists.

It underlines the result brought to us by the LACC study for the early cervical cancer which showed the inferiority of minimally invasive surgery in the management of early cervical cancer in terms of disease-free survival (DFS) and overall survival (OS) compared to laparotomy. These findings have radically changed our practice, so much so that ESGO and NCCN have modified their recommendations in favor of laparotomy in the surgical management of  early cervical cancer.

The study can be accepted after a minor revision

Authors refer to “early cervical cancer” but IB2 is not considered early-stage disease.

Early cervical cancer refers to FIGO IA1-IB1, while IB2 belong to locally advanced cervical cancer.

Results

- Line 69: IB2 is not considered “early cervical cancer” so I would remove it.

-Line 218: In the LACC study patients were eligible if they had a disease stage of IA1, IA2 or IB1 (not IB2).

The drafting of the results is not very linear, I would have added more explanatory tables with authors, stage of the disease and results of the study.

Author Response

September 25, 2023

Dear Editors,

We received the reviewer’s comments regarding our manuscript submitted for consideration for publication entitled "Surgical treatment for Early Cervical Cancer in the HPV era: state of the artand we are grateful for the opportunity to revise our work. We would like to thank the reviewers for considering the work interesting and for taking the time to make those very appropriate comments to improve it. We have modified the text of the manuscript to address all the areas identified by the reviewers. We believe these revisions have adequately addressed the raised points, and we hope that the revised version of the manuscript is now considered acceptable for publication.

ACCORDING TO REVIEWER 4’S SUGGESTIONS:

The study emphasizes concepts significant to oncology gynecologists.

It underlines the result brought to us by the LACC study for the early cervical cancer which showed the inferiority of minimally invasive surgery in the management of early cervical cancer in terms of disease-free survival (DFS) and overall survival (OS) compared to laparotomy. These findings have radically changed our practice, so much so that ESGO and NCCN have modified their recommendations in favor of laparotomy in the surgical management of  early cervical cancer.

The study can be accepted after a minor revision.

  • “Authors refer to “early cervical cancer” but IB2 is not considered early-stage disease. Early cervical cancer refers to FIGO IA1-IB1, while IB2 belong to locally advanced cervical cancer”.
  • We have corrected it, in the following points as you suggested.

Results

  1. Line 69: IB2 is not considered “early cervical cancer” so I would remove it.
  2. We removed the words “in early stages” (69 line).
  3. Line 218: In the LACC study patients were eligible if they had a disease stage of IA1, IA2 or IB1 (not IB2).
  4. We corrected it as requested, We apologize for the error (218 line).
  5. The drafting of the results is not very linear, I would have added more explanatory tables with authors, stage of the disease and results of the study”.
  6. Thanks for the advice, I have added a new table in this regard which quickly summarizes the points of the works that have changed the way of thinking about the treatment of this tumor (table 4).

******

We agree with the reviewer. All the points highlighted by the reviewer represent the future aims of our research line. Our study represented a promising starting point and showed that this approach could discriminate between patients affected by endometriosis and not.  However, our study shows several weak points, including those highlighted by the reviewer.

We elaborated the conclusion section describing strengths e weakness points of the paper including those mentioned by the reviewer.

Once again, we would like to thank the Reviewers for the precious suggestions and the Editor for allowing us to improve our manuscript. We hope our paper now has the quality to be accepted for publication in your prestigious Journal.

We remain at your disposal for any further detail you might consider essential to clarify.

On behalf of the co-Authors,                                                                   

Luigi Della Corte*

*[email protected]; [email protected] (Luigi Della Corte, M.D. 1Department of Neuroscience, Reproductive Sciences and Dentistry, School of Medicine, University of Naples “Federico II”, 80131 Naples, Italy. ORCID: 0000-0002-0584-2181.)

Round 2

Reviewer 2 Report

Dear Authors 

The present paper proposes an overview of the surgical approach to early-stage cervical cancer, especially regarding to the LACC study depending on certain criteria and the surgical procedure.

The changes made to the section related to screening tests bring additional information related to their usefulness and the expression has been improved.

Also, the Discussion section was revised so that the analysis made on the data from the literature was deepened and the comparison with the current study made a more detailed analysis.

In conclusion, the changes made to the current work bring more accuracy, which makes the work worth publishing in its current form.

Kind regards

Author Response

I want to thank you on behalf of myself and all the authors for your support and for the opportunity you gave us to improve the manuscript.
